# NEURON TO GRAPH: INTERPRETING LANGUAGE MODEL NEURONS AT SCALE

## ABSTRACT

Advances in Large Language Models (LLMs) have led to remarkable capabilities, yet their inner mechanisms remain largely unknown. To understand these models, we need to unravel the functions of individual neurons and their contribution to the network. This paper introduces a novel automated approach designed to scale interpretability techniques across a vast array of neurons within LLMs, to make them more interpretable and ultimately safe. Conventional methods require examination of examples with strong neuron activation and manual identification of patterns to decipher the concepts a neuron responds to. We propose Neuron to Graph (N2G), an innovative tool that automatically extracts a neuron's behaviour from the dataset it was trained on and translates it into an interpretable graph. N2G uses truncation and saliency methods to emphasise only the most pertinent tokens to a neuron while enriching dataset examples with diverse samples to better encompass the full spectrum of neuron behaviour. These graphs can be visualised to aid researchers' manual interpretation, and can generate token activations on text for automatic validation by comparison with the neuron's ground truth activations, which we use to show that the model is better at predicting neuron activation than two baseline methods. We also demonstrate how the generated graph representations can be flexibly used to facilitate further automation of interpretability research, by searching for neurons with particular properties, or programmatically comparing neurons to each other to identify similar neurons. Our method easily scales to build graph representations for all neurons in a 6-layer Transformer model using a single Tesla T4 GPU, allowing for wide usability.

## 1 INTRODUCTION

Interpretability of machine learning models is an active research topic (Hendrycks et al., 2021; Amodei et al., 2016) and can have a wide range of applications from bias detection (Vig et al., 2020) to autonomous vehicles (Barez et al., 2022) and Large Language Models (LLMs; (Elhage et al., 2022a)). The growing sub-field of mechanistic interpretability aims to understand the behaviour of individual neurons within models as well as how they combine into larger circuits of neurons that perform a particular function (Olah et al., 2020b; Olah, 2022; Goh et al., 2021), with the ultimate aim of decomposing a model into interpretable components and using this to ensure model safety.

Feature visualisation (Olah et al., 2017) is a tool for interpreting neurons in image models, whereby a synthetic input image is optimised to understand a target neuron. This has significantly aided work interpreting vision models. For example, to identify multimodal neurons which respond to abstract concepts (Goh et al., 2021), and to catalogue the behaviour of all early neurons in Inceptionv1 (Olah et al., 2020a). Similar interpretability tools for understanding neurons in LLMs are lacking. Currently, researchers often look at dataset examples containing tokens upon which a neuron strongly activates and investigate common elements and themes across examples to give some insight into neuron behaviour (Elhage et al., 2022a; Geva et al., 2020). However, this can give the illusion of interpretability when real behaviour is more complex (Bolukbasi et al., 2021a), and measuring the degree to which these insights are correct is challenging. Additionally, inspecting individual neurons by hand is time-consuming and unlikely to scale to entire models.

To overcome these challenges, we present **Neuron to Graph (N2G)**, which automatically converts a target neuron within an LLM to an interpretable graph that captures a neuron's behaviour. Our

method takes maximally activating dataset examples for a target neuron, prunes them to remove irrelevant context, identifies the tokens which are important for neuron activation, and creates additional examples by replacing the important tokens with likely substitutes using DistilBERT (Sanh et al., 2020). These processed examples are then given as input to the graph builder, which removes unimportant tokens and creates a condensed graph representation. The graph can be visualised to facilitate understanding the neuron's behaviour, as well as used to process text and produce predicted token activations. This allows us to measure the correspondence between the target neuron's activations and the graph's structure, which provides a direct measurement of the degree to which a graph captures the neuron's behaviour. Once built, the graphs can be searched to quickly identify neurons with particular properties, which could help facilitate interpretability research.

N2G provides a promising research direction for understanding the function of individual neurons in language models by building interpretable representations for every neuron in a model. Our main **contributions** are four-fold:

- An input pruning method which removes any context from an input example that is unnecessary for neuron activation.
- Token saliency computation to identify the importance of each context token for neuron activation.
- An augmentation technique for pruned inputs to better explore and understand neuron behaviour by generating varied inputs through predicted token substitutions.
- A graph-building process that results in a graph representation for each neuron. The quality of the representation can be automatically measured by comparing the output of the representation to the real neuron, the graph can be visualised for human interpretation, and the representations can be searched and compared to help automate parts of interpretability research.

## 2 RELATED WORK

Neuron analysis is a branch of natural language processing (NLP) research that investigates the structure and function of neurons within an LLM. It plays an important role in understanding the inner workings of a model and has the potential to enable a mechanistic understanding of large models.

Prior work in neuron analysis has identified the presence of neurons correlated with specific concepts (Radford et al., 2017b). For instance, (Dalvi et al., 2019) explored neurons which specialised in linguistic and non-linguistic concepts in large language models, and (Seyffarth et al., 2021) evaluated neurons which handle concepts such as causation in language. The existence of similar concepts embedded within models can also be found across different architectures. (Wu et al., 2020) and (Schubert et al., 2021) examined neuron distributions across models and found that different architectures have similar localised representations of information, even when (Durrani et al., 2020) used a combination of neuron analysis and visualisation techniques to compare transformer and recurrent models, finding that the transformer produces fewer neurons but exhibits stronger dynamics.

There are various methods of identifying concept neurons (Geva et al., 2020). In (Bau et al., 2018), a method was proposed for identifying important neurons across models by analyzing correlations between neurons from different models. In contrast, (Dai et al., 2021) developed a method to identify concept neurons in transformer feed-forward networks by computing the contribution of each neuron to the knowledge prediction. In contrast, we focus on identifying neurons using highly activating dataset examples. (Mu & Andreas, 2020) demonstrated how the co-variance of neuron activations on a dataset can be used to distinguish neurons that are related to a particular concept. (Torroba Hennigen et al., 2020) also used neuron activations to train a probe which automatically evaluates language models for neurons correlated to linguistic concepts.

One limitation of using highly activating dataset examples is that the accurate identification of concepts correlated with a neuron is limited by the dataset itself. A neuron may represent several concepts, and (Bolukbasi et al., 2021b) emphasise the importance of conducting interpretability research on varied datasets, in order to avoid the "interpretability illusion", in which neurons that show consistent patterns of activation in one dataset activate on different concepts in another. (Poerner et al., 2018) also showed the limitations of datasets in concept neuron identification. They demonstrated that generating synthetic language inputs that maximise the activations of a neuron surpasses naive search on a corpus.

Researchers also used GPT-4 to simulate neurons and predict neuron activation, and then again use it to explain neuron behaviour (Bills et al., 2023). This method represents an important step towards scalable neuron interpretability, providing the ability to automatically generate explanations for all neurons in a Language Model. However, whilst a natural language explanation can be very useful, it does not enable further automated analysis in the same way a more structured representation can. For example, searching explanations requires complex semantic search compared to simple syntactic matching, and similarly comparing the explanations for multiple neurons is more challenging.

## 3 METHODOLOGY

In this section, we first discuss the model architecture we seek to create neuron interpretations for (§3.1) and then discuss our algorithm for creating these interpretations (§3.2).

### 3.1 MODEL ARCHITECTURE

To test our algorithm, we analyse neurons of a SoLU model Elhage et al. (2022a), an auto-regressive model from the Transformer family. This model uses the SoLU activation function, which pushes neurons to be monosemantic by penalising many neurons in a layer from firing at once. SoLU models have been shown to have a higher prevalence of neurons that represent a single feature, and are therefore more easily interpretable. They therefore provide an ideal test-bed for our model.

Similarly to other auto-regressive Transformer models, the SoLU model implementation we use (Nanda, 2022) includes multi-layered perceptron (MLP) layers within the Transformer blocks that make up the model. The model has six blocks, each with one MLP layer containing 3072 neurons, and was trained on the Pile (Gao et al., 2020). We denote the set of neurons from all MLP layers by $\mathcal{N}$, where each element in $\mathcal{N}$ is $n_{\ell j}$, indexing the $j$th neuron in layer $\ell$ of the SoLU model.

During its feed-forward step on a set of tokens, $x = x_1 \cdots x_n$, a given neuron $n_{\ell j}$ generates an activation over all tokens $x_1 \cdots x_n$. We denote by $a(i, x, \ell, j)$ the function that returns the activation of neuron $j$ in layer $\ell$ on input $x$ for the $i$th token $x_i$. We denote by $a(\cdot, x, \ell, j)$ to be $a(|x|, x, \ell, j)$. For integers $c, d$, we denote by $x_{c:d}$ the substring $x_c x_{c+1} \cdots x_d$.

Given a fixed neuron $n_{\ell j} \in \mathcal{N}$ that we are seeking to create an interpretation for, we obtain the maximum activating dataset sample from the Neuroscope resource (Nanda, 2022). This resource provides the top 20 most activating examples from the training dataset for each neuron in the model. Each dataset example is a 1024-token string, and we refer to the final activating dataset sample of strings for any particular neuron as $\mathcal{X} = \{x^{(1)}, \ldots, x^{(m)}\}$.

### 3.2 N2G: NEURON TO GRAPH ALGORITHM

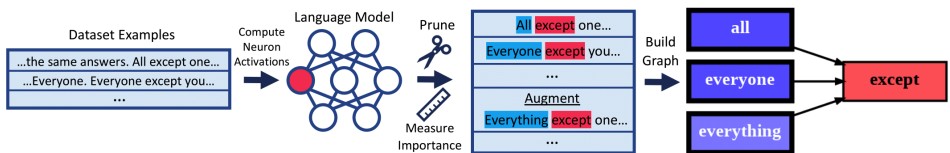

Figure 1: **Overall architecture of N2G.** Activations of the target neuron on the dataset examples are retrieved (neuron and activating tokens in red). Prompts are pruned and the importance of each token for neuron activation is measured (important tokens in blue). Pruned prompts are augmented by replacing important tokens with high-probability substitutes using DistilBERT. The augmented set of prompts are converted to a graph. The output graph is a real example which activates on the token "except" when preceded by any of the other tokens.

For each neuron in the model we automatically build an interpretable representation that aims to capture the target neuron's behaviour. Our algorithm is inspired by the notion that the role of a neuron, and hence its possible interpretation, can be understood by examining the cases in which its activation levels are high. We, therefore, aim to expand the list of sequences in $\mathcal{X}$ to related sequences that also generate high activations for the target neuron $n_{\ell j}$. This expanded list of sequences

is then compiled into a lattice (a form of a directed acyclic graph) where each path in this lattice is a *minimally* representative token sequence which results in high activation for the neuron. We present pseudocode for the algorithm in the Appendix.

**Pruning**

The algorithm begins with a pruning step, which aims to extract the key substring of each maximally activating dataset example. More formally, for each $x^{(i)} \in \mathcal{X}$ it identifies the pivot token index:

$$e_i = \arg \max_k a(k, x^{(i)}, \ell, j). \tag{1}$$

This pivot token has the highest activation among all tokens in $x^{(i)}$ for the target neuron $n_{\ell j}$, and therefore, is assumed to be the representative token for interpreting $n_{\ell j}$. Given this pivot token, we want to find the shortest prior context required to activate the neuron. Following the identification of $e_i$, we find a minimal subsequence of $x^{(i)}$ that ends in position $e_i$ and starts in position $s_i$ such that:

$$\frac{a(e_i, x_{s_i}^{(i)} \cdots x_{e_i}^{(i)}, \ell, j)}{a(e_i, x^{(i)}, \ell, j)} \geq 0.5. \tag{2}$$

Concretely, we iteratively add tokens to the left of the pivot token, each time re-calculating the activation on the pivot token in the new subsequence until it is at least half (using empirical guidance) of the activation on the pivot token in the full sequence. The purpose of this pruning is to remove extraneous information that is irrelevant for neuron activation, which facilitates the later stages of the algorithm. After pruning, we end with a set of minimal subsequences $\mathcal{Y} = \{y^{(1)}, \ldots, y^{(m)}\}$ where each one is generated from a single element in $\mathcal{X}$, $y^{(i)} = x_{s_i: \, e_i}^{(i)}$.

**Saliency Identification**

Given the set $\mathcal{Y}$ of strings, we follow another saliency identification step. For each example $y^{(i)} \in \mathcal{Y}$, we measure the relative importance of the $k$th token $y_k^{(i)}$ to the activation of the target neuron $n_{\ell j}$ on the $k'$th token $x_{k'}$. We calculate the value:

$$\alpha_{k, k'} = 1 - \frac{a(k, \hat{y}^{(i)}, \ell, j)}{a(k', y^{(i)}, \ell, j)}, \tag{3}$$

where in this context, $\hat{y}^{(i)}$ is the same as the sequence $y^{(i)}$, except that the $k$th token is replaced with a special padding token. The relative importance $\alpha_{k, l}$ indicates what would happen to the target neuron activation on the $k'$th token of $y^{(i)}$ if we perturbed the $k$th token of that sequence. Intuitively, if the activation of the neuron on token $k'$ is much lower in the perturbed sequence than the original sequence, the token $k$ is providing necessary context that is important for neuron activation. This information is useful in later stages of the algorithm, allowing us to identify the key context tokens for a given neuron, and discard the unimportant tokens that are irrelevant to the neuron's behaviour.

**Augmentation**

We then automatically augment our set of examples $\mathcal{Y}$ with additional ones. For each example $y^{(i)} \in \mathcal{Y}$, we identify the tokens that are important for neuron activation on the pivot token by thresholding the value $\alpha_{k, e_i}$ for all tokens at index $k < e_i$. Each of these important tokens is then replaced with related tokens that are predicted from a helper model, such as BERT (Devlin et al., 2018). These tokens, $\mathcal{R}_{k, i}$ are a set of elements in the vocabulary that the helper model gives high probability to at position $k$ when fed with $y^{(i)}$ masked at position $k$. Once we have generated a candidate set of augmentations $\mathcal{A}$, we pass each new example $a^{(i)}$ through the target model and measure the activation of the target neuron on the pivot token, $a(e_i, a^{(i)}, \ell, j)$, and enrich $\mathcal{Y}$ with all $a^{(i)}$ where that pivot activation is more than half of the pivot activation on the originating sequence $y^{(i)}$:

$$\frac{a(e_i, a^{(i)}, \ell, j)}{a(e_i, y^{(i)}, \ell, j)} \geq 0.5. \tag{4}$$

Intuitively, we are exploring the space around each input example to find other, similar, examples, and retaining the ones that still strongly activate the target neuron. This allows us to better understand the full extent of a neuron's behaviour. We now update $\mathcal{Y}$ to include all augmented examples.

For our helper model, we chose DistilBERT (Sanh et al., 2020), as its bidirectionality provides the ability to use both the prior and post context for better token predictions. In addition, DistilBERT is a compact and efficient variant of the BERT model (Devlin et al., 2018), trained using knowledge distillation, allowing it to run 60% faster than BERT with similar statistical performance. This makes it an ideal choice for efficiently predicting substitute tokens.

**Graph Building**

After completing the previous stages, we have a set of dataset examples $\mathcal{Y}$ which strongly activate our target neuron. We then aim to build a compact representation which concisely represents the tokens on which our neuron activates, as well as the context necessary for activation on these tokens. To do this, we create a lattice structure which has tokens as nodes, where each path is constructed from a sequence in $\mathcal{Y}$ and contains the important context tokens for activation on the final pivot token, which ends the path.

For each example $y \in \mathcal{Y}$, we iterate over the tokens and retrieve the importance of that token relative to the pivot token. For each token $y_k$, where $k < e_i$, if its importance is above the threshold we add it to the path as an important token. Otherwise, we add it to the path as a special ignore token. We then end the path with pivot token. Ignore tokens are placeholder's indicating that some token is necessary here, but that the specific token does not matter - in contrast to an important token, where that specific token has been identified as key for neuron activation the pivot token.

After creating the paths, we store them in a trie. We work backwards through each path, first adding the pivot token as a top layer **activating** token, then adding each **context** token as an important node or an ignore node as required. At the end of each path we add a special end node, which denotes a complete path through the trie. On this end node, we store the normalised activation of the target neuron on the activating token for that path. We compute the normalised activation by retrieving the maximum observed activation of the neuron on any token in the training dataset $\mathcal{D}$, $a_{max} = \max_{i,k} a(k, x^{(i)}, \ell, j)$. We then compute the normalised activation as $a(e_i, y, \ell, j)/a_{max}$ for the example $y$ which formed the path.

By storing our representation in this way, we can use it to simulate neuron behaviour by predicting token activations. Specifically, we can pass input tokens $x = x_1 \cdots x_n$ to the trie, and it will output a predicted normalised activation for each token $x_i$ in $x$. Starting at the root of the trie, we check if the token $x_i$ is in the root's child nodes. If it is, we have identified that $x_i$ is an activating token, so we then continue to check if there is the necessary context for activation. Iterating backwards starting at $x_{i-1}$, we check if the token matches any of the current node's children - specifically, whether the token is either in the current node's children, or whether the current node has an ignore token in its children. If at any point we reach a node that has an end node as a child, we have traced a valid path through the trie, so record the normalised activation stored on the node. We continue these steps until we run out of prior tokens, or the current token fails either of the matching conditions. We then return the maximum observed normalised activation, or 0 if we did not find any matching paths.

This representation therefore allows us to directly measure the correspondence between a target neuron's activations and the predicted activations from the trie for any input text, which allows us to quantify the accuracy of the representation. In addition, we are able to visualise the representation to facilitate human interpretation. To do this, we remove ignore nodes as they do not contain relevant information, then at each layer in the trie we collapse identical tokens to form the directed graph structure. We colour context nodes as blue according to their importance, and activating nodes as red according to their normalised activation. Brighter colours indicate higher importance or activation, respectively. Additionally, we denote valid end nodes with a bold outline.

We refer to this visualisation, as well as the underlying trie, as a **neuron graph**, which we can visually inspect or automatically evaluate. Figure 1 shows an example of a simple graph for a neuron

in layer 1 of the model. The neuron graphs also facilitate new forms of analysis. For example, they provide a simple searchable structure. We provide the ability to efficiently search graphs by both activating tokens and context tokens. For example, we could search for a graph that activates on the token "everyone" when it occurs with the context token "except", to retrieve the graph in Figure 1. This search functionality allows us to quickly look for neurons with a particular behaviour in which we are interested. In comparison, attempting to search maximally activating dataset examples in the same way is much less useful, as they often contain significant amounts of unimportant information which renders any search on context tokens useless.

## 4   RESULTS AND DISCUSSION

We describe a quantitative analysis, comparing our methods to simple look-up methods (§4.1) and two case studies where we use N2G to discover interesting neuron behaviour (§4.2). We also provide additional case studies in the Appendix.

### 4.1   QUANTITATIVE RESULTS

Given that the neuron graphs built by the algorithm can be directly used to process text and predict token activations, we can evaluate the degree to which they accurately capture the target neuron's behaviour by measuring the correspondence between the activations of the neuron and the predicted activations of the graph on some evaluation text. We conduct all experiments on the SoLU model described in §3.1.

For each neuron, we take the top 20 dataset examples from Neuroscope and randomly split them in half to form a train and test set. We give the training examples to N2G to create a neuron graph, then take the test examples compute the normalised token activations as described in §3.2. We apply a threshold to the normalised token activations, defining an activation above the threshold as a *firing* of the neuron, and an activation below the threshold as the neuron *not firing*. In these experiments we set the threshold to $0.5$. We then process the test prompts with the neuron graph to produce predicted token firings. We can then measure the precision, recall, and $F1$ score of the graph's predictions compared to the ground truth firings. Building a neuron graph for every neuron in the model took approximately 48 hours of processing on a single Tesla T4 GPU.

To ground the results of N2G we compare to two simple baselines. The first is a per-neuron token lookup table. For each neuron, we take the same training set of 10 dataset examples which we give to the N2G algorithm, pass them through the model and record the activation of the neuron on each token. We then create a lookup table that maps every token to the maximum observed activation for any occurrence of that token. We can then predict token activations on text by outputting the stored activation for each token in the input (or 0 if we did not see the token when creating the lookup). Intuitively, this presents a very recall-focused baseline, as it ignores any context and instead just identifies whether a token has ever been seen to activate a neuron. The second baseline generalises the token lookup to an n-gram lookup, following the same process as above but storing the prior $n$ context tokens for each token in the input. We choose $n = 5$ for our baseline, which provides a precision focused baseline that requires specific context for neuron activation on any given token.

Table 1 shows the average precision, recall, and $F1$ score of the three methods for each layer of the model, averaged across the neurons in that layer For each neuron, we only compute these statistics on the tokens that caused the neuron to fire. This is because any given neuron typically fires on very few tokens, so the prediction problem is very imbalanced - predicting the neuron will never fire would give very good performance but provide no useful information.

In layers 0 and 1 of the model, the neuron graphs built with N2G on average capture the behaviour of the neurons well, with high recall and decent precision on the firing tokens. In comparison to the baselines, N2G achieves close to the recall of the high recall baseline, with substantially better precision. However, the precision baseline achieves significantly higher precision than N2G, suggesting that the method is not fully capturing the context needed to precisely determine when the neuron will fire.

However, as we progress to deeper layers of the model, the recall and precision of N2G generally decreases. This corresponds to neurons in the later layers on average exhibiting more complex

| | N2G | | | Token Lookup | | | n-Gram Lookup | | |
|---|---|---|---|---|---|---|---|---|---|
| Layer | Precision | Recall | F1 | Precision | Recall | F1 | Precision | Recall | F1 |
| 0 | 0.67 | 0.84 | **0.68** | 0.45 | 0.89 | 0.53 | 0.97 | 0.13 | 0.17 |
| 1 | 0.64 | 0.82 | **0.65** | 0.36 | 0.90 | 0.44 | 0.96 | 0.14 | 0.18 |
| 2 | 0.56 | 0.75 | **0.55** | 0.29 | 0.85 | 0.37 | 0.93 | 0.14 | 0.17 |
| 3 | 0.45 | 0.71 | **0.45** | 0.23 | 0.82 | 0.30 | 0.91 | 0.16 | 0.18 |
| 4 | 0.39 | 0.66 | **0.38** | 0.22 | 0.78 | 0.28 | 0.89 | 0.16 | 0.18 |
| 5 | 0.38 | 0.68 | **0.37** | 0.20 | 0.81 | 0.27 | 0.86 | 0.26 | 0.28 |

Table 1: N2G compared to two baseline. Note the statistics are averaged across all neurons in a layer, and only computed for the tokens on which the neuron fired. Best $F1$ scores in bold.

behaviour that is less completely captured in the training examples. Specifically, neurons in early layers tend to respond to a small number of specific tokens in specific, narrow contexts, whereas later layers often respond to more abstract concepts represented by a wider array of tokens in many different contexts, which was similarly observed by (Elhage et al., 2022a). We see similar trends for the recall baseline, and the decrease in recall in particular suggests that the training examples on which we fit our models do not fully capture the behaviour of neurons in later layers. This suggests there could be improvements from collecting a larger set of dataset examples for each neuron and demonstrates the value of augmentation to better explore the space of activating inputs.

Interestingly, the recall of the n-gram lookup increases as we go deeper into the model. This is likely because later layers tend to require longer context and more specific context, whereas early layers may only require very short context, less than the 5 tokens used in the baseline. This demonstrates the importance of the pruning and saliency mechanisms, to adaptively identify the necessary context for activation on a per neuron basis.

The trend of later layers requiring more context for activation is demonstrated by Figure 2 a). This shows how the normalised activation of a neuron on the pivot token in an input changes as we progressively remove tokens from the prior context. We refer to this as an **activation trajectory**, and compute the average trajectory across 100 neurons in each layer of the model. We observe that as we move from early to later layers the average activation trajectory becomes shallower, showing that neurons in later layers on average require a much longer context to activate than neurons in early layers.

We show a typical example of a neuron from layer 4 of the model in Figure 2 (b). The average trajectories show a smooth decrease in activation as we remove context tokens, but the more typical shape of any individual neuron is for there to be distinct jumps in activation as we reach important tokens. The trajectories start off identically for the first five tokens corresponding to "category: reptiles", which are important for activation and therefore included in the neuron graph at the bottom of the figure. There is then a longer-range dependency on the token "idae" (which for example, could occur at the end of the word "Gekkonidae", the scientific name for Geckos), which can be

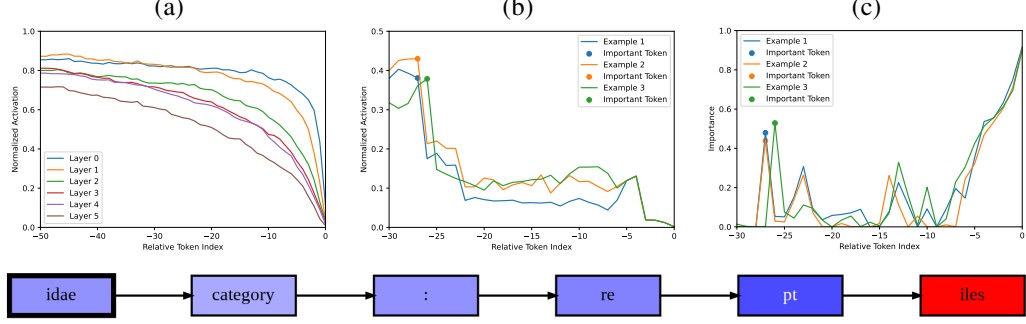

Figure 2: (a) Activation trajectory averaged across 100 neurons from each layer of the model. (b) and (c): Activation and Importance trajectories for a neuron from layer 4. Points show occurrence of the important "idae" token. Below, graph for the corresponding neuron from layer 4.

of variable length. Distinct spikes occur with this token, both in the activation and importance trajectories, marked with points in the graphs. This demonstrates how the neuron graph captures the key information for activation, whilst abstracting away information about the length dependency between tokens. We also see that the importance measure from the saliency mechanism closely relates to the activation trajectories, with a spike in importance corresponding to the increase in activation.

## 4.2 LARGE SCALE NEURON CASE STUDIES

Constructing the neuron graphs for every neuron in a model enables new workflows for mechanistic interpretability research. They provide a simple searchable structure which we can use to identify neurons with specific properties, and can be compared to each other to find similarities between neurons. Here, we present some case studies demonstrating how these representations can enable these forms of investigation.

**In-context Learning**

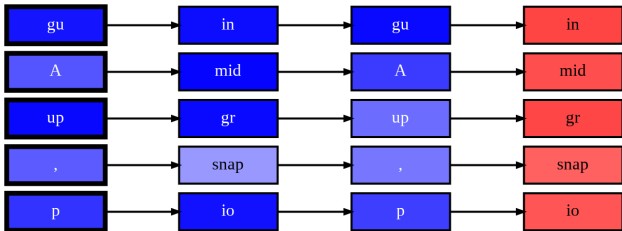

Figure 3: Neuron graph for an in-context learning neuron that activates on repeated token sequences.

In-context learning refers to the ability of Language Models to use prior information in their prompt to better predict later tokens, and is a defining property of these models. A key mechanism behind in-context learning is the induction head (Olsson et al., 2022), an attention head which learns to look for repeated token sequences and increase the probability of predicting later tokens in the sequence if the beginning of the sequence is seen again. For example, if we see "abc...ab", an induction head would contribute to increasing the probability of predicting the token "c" will reoccur.

We automatically searched the graph representations to identify graphs that contained any particular token in both the context tokens and the activating tokens. This enabled us to discover many neurons that activate on repeated tokens, and in particular we found what appears to be an in-context learning neuron. Shown in Figure 3, this neuron responds to a wide variety of apparently unrelated tokens, as long as they are part of a repeating sequence such as "ab...a**b**". This demonstrates how the neuron graph representation can enable researchers to much more efficiently explore and understand particular neuron behaviours via search, facilitating mechanistic interpretability research.

**Discovering Neuron Groups**



Figure 4: A neuron graph that occurs for a neuron in Layer 1 and a neuron in Layer 4 of the model. The neurons have identical behaviour, and were recognised as a similar pair through an automated graph comparison process.

In addition, this representation offers the possibility of automatically identifying higher level structures within Language Models, such as simple circuits (Elhage et al., 2021). We provide an initial step towards this by automatically identifying neurons with similar behaviours by comparing their representations. Specifically, for every pair of neurons we measure the proportion overlap between the pairs' context tokens, as well as the proportion overlap between their activating tokens. We can then automatically identify similar neurons by retrieving pairs with more than 90% overlap for both context tokens and activating tokens.

We identify 60 pairs of similar neurons within the model by automatically comparing pairs of neuron graphs. Figure 4 shows an example of a neuron graph that represents a neuron in layer 1 of the model. Our analysis identified that an identical graph also represents a neuron in layer 4 of the model.

A possible explanation for the existence of such pairs of neurons is the spreading out of features due to superposition. Previous work has provided evidence for superposition, where models use individual neurons to represent multiple unrelated features. This phenomenon is a barrier to interpretability as it makes understanding neuron function more challenging (Elhage et al., 2022a). Superposition implies that features are spread across multiple neurons (Olah et al., 2020b), to enable a model to represent more features than it has neurons. As such, it could provide an explanation for the occurrence of similar neurons. Alternatively, the model may be incentivized by dropout during the training process to represent highly important features in multiple neurons, so that dropping out any particular neuron does not hurt the loss. Further research into superposition and the existence of neurons with similar behaviours would be beneficial to better understand the causes of these phenomena.

Superposition also has implications for attempting to control undesirable behaviours. One method for preventing a model from exhibiting some negative behaviour, such as toxic outputs, would be to identify specific neurons that are responsible for this behaviour and ablate them. Indeed, (Radford et al., 2017a) found such a sentiment neuron in an LSTM model, and used it to control the sentiment of generated text. Superposition implies that features such as sentiment or toxicity may be spread across multiple neurons (Elhage et al., 2022a), so ablating any a single neuron may not significantly affect the feature representation. The ability to identify similar neurons therefore could facilitate identifying clusters of neurons that contribute to a negative behaviour, and allow us to efficiently ablate all necessary neurons. This shows the potential for the N2G representations to facilitate diverse tasks in ML safety.

## 5 CONCLUSIONS AND LIMITATIONS

We presented N2G, an approach for converting neurons in LLMs into interpretable graphs that can be visualised, evaluated and searched. The degree to which a neuron graph captures the behaviour of a target neuron can be directly measured by comparing the output of the graph to the activations of the neuron, making this method a step towards scalable interpretability methods for LLMs. We find that neuron graphs capture neuron behaviour well for early layers of the model but only partially capture the behaviour for later layers due to increasingly complex neuron behaviour, and this problem will likely become more prominent in larger models. Additionally, we evaluate our method on a SoLU (Elhage et al., 2022a) model to minimise polysemanticity, though it can also be applied to models with more common activation functions. The greater prevalence of polysemantic neurons in typical Transformer models could reduce the ability of N2G to fully capture neuron behaviour. Future work could address these limitations by using more training examples to better cover the full extent of a neuron's behaviour, better exploring the input space via more sophisticated and extensive augmentation, and generalising from exact token matches to matching abstract concepts, for example, by using token embeddings.

The neuron graphs built by N2G allow for new methods of analysis. They can be searched to identify neurons with interesting properties, which we demonstrate by finding an in-context learning neuron that responds to repeated sequences of the form "ab...ab". They can also be programmatically compared with each other, which we use to identify pairs of neurons with very similar behaviours. These case studies show how the graph representation can enable new forms of interpretability analysis and make some tasks much faster by automating parts of the discovery process.

There could be significant scope to develop new processes that act on the neuron representations to assist with interpretability research. For example, future work could look at using the graph representations to automatically identify circuits of neurons within LLMs that together perform a particular task. For example, we could automatically look for instances where multiple neurons that each respond to a single concept are combined to form a neuron in a later layer that responds to all of those concepts, by extending the similarity comparison to identify neurons which have a subset of another neuron's behaviour. In general, once we have generated the neuron graphs for all neurons in a model, these become a valuable resource upon which researchers can develop new tools for analysis.

## 6 REPRODUCIBILITY STATEMENT

To ensure our work is reproducible, we provide the full source code in the supplementary materials, as well as all necessary data and parameters and instructions to reproduce our experimental results.

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

## A    NEURON GRAPH EXAMPLES

In this appendix we explore some interesting or characteristic behaviours of the neuron graphs. Polysemanticity, the phenomenon where a neuron exhibits multiple unrelated behaviours, is one of the current major challenges of neuron interpretability (Elhage et al., 2022b). When present, polysemanticity often shows up clearly in the neuron graphs as distinct, disconnected subgraphs. For example, in Figure 5, there are three separate subgraphs corresponding to three clearly distinct behaviours. The top subgraph responds to a phrase in Dutch - variations on *de betrokken*, where not all tokens in *betrokken* were important enough to include in the graph. The middle subgraph responds to a phrase in English - variations on *a fun, over the top*. The bottom subgraph responds to a phrase in Swedish - *kollegers berättigade*, with unimportant tokens not included. This natural separation of behaviours into separate subgraphs could potentially make it easier to interpret polysemantic neurons, but more experimentation would be needed to develop and test this further.

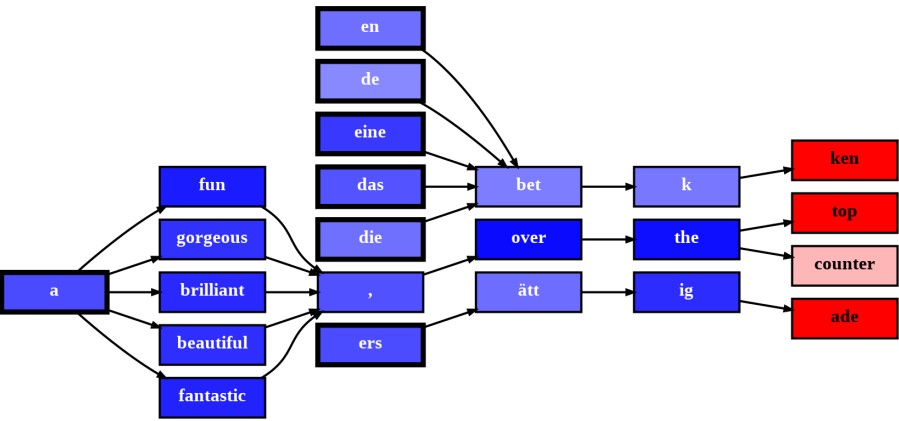

Figure 5: A neuron graph exhibiting polysemanticity, with three disconnected subgraphs each responding to a phrase in a different language.

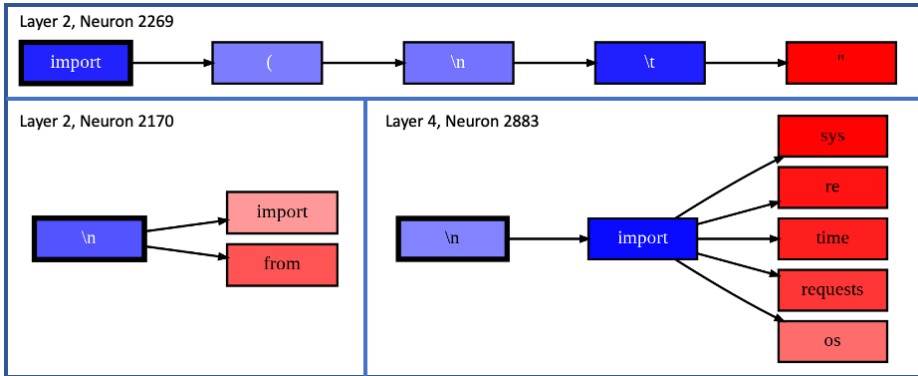

Figure 6: Neurons related to programming syntax, specifically import statements. Top - Neuron graph illustrating import syntax for the Go programming language. Bottom Left: Neuron graph showing fundamental elements of Python import syntax. Bottom Right: Neuron graph for a neuron that responds to the imports of widely-used Python packages.

Neuron graphs appear to work particularly well for "syntactic neurons" that respond to structural patterns, concisely and simply capturing the structure. One rich source of syntactic text is programming, suggesting that N2G could particularly useful for analysing models trained to write code. We use the search capability on the neuron graphs of the SoLU model to identify many neurons related to import statements in various languages. Figure 6 shows three examples from different layers of the model. The top and bottom left are from layer 2, and represent basic syntax in Go and Python respectively. The bottom right graph is for a neuron in layer 4, and shows a neuron that responds to imports of common Python packages. The circuits line of research (Olah et al., 2020b) would suggest that later-layer neurons like the one in layer 4 may be "composed" of neurons in early layers - for example, a simple way to do this would be to union over several neurons that each respond to an import of one of the packages (i.e., the later composed neuron activates if any of the previous neurons activates). Moving from understanding individual neurons to understanding circuits of neurons is a crucial step in interpretability research. The ability to automatically identify similar neurons could be expanded to identify neurons that have a subset of another neuron's behaviour, which could provide a method for discovering simple circuits in language models. This demonstrates how the flexible representations built by N2G could help facilitate new methods for interpretability research.

## B  N2G PSEUDOCODE

---

**Algorithm 1**

---

1: **procedure** N2G ALGORITHM

   **Input**: Target neuron $n_{\ell j}$, list of sequences $\mathcal{X}$

   **Prune, Saliency Identification and Augment Steps:**

2:   **for** $x^{(i)} \in \mathcal{X}$ **do**

3:     Compute $e_i \leftarrow$ pivot token index

4:     Find $y^{(i)} \leftarrow$ minimal sub sequence with activation ratio $\geq 0.5$

5:     Form $\mathcal{Y} \leftarrow \{y^{(1)}, \ldots, y^{(m)}\}$

6:   **for** $y^{(i)} \in \mathcal{Y}$ **do**

7:     Compute relative importance value $\alpha_{i,k}$ for each token in $y_k^{(i)}$

8:     Identify tokens with high $\alpha_{i,k}$

9:     Obtain replacement tokens $\mathcal{R}_{k,i}$ from helper model

   **Construct Lattice and Optimise Steps:**

10:    Combine $y^{(i)}$ and $\mathcal{R}_{k,i}$ to form a lattice of augmented minimal subsequences

11:    Find optimal token combination in the lattice that maximises target neuron activation $n_{\ell j}$

   **Output**:

12:    The optimal lattice of augmented minimal subsequences representing high activation contexts for the target neuron $n_{\ell j}$

---

## C    BROADER IMPACT

It is worth mentioning that the field of mechanistic interpretability is ideal for understanding the black box nature of neural networks. This approach can help improve model comprehension and enable researchers to build more transparent AI systems. However, it is essential to recognize that as models become more capable, they may also be used for purposes that are not aligned with societal needs and safety. This presents potential ethical concerns and underscores the need for responsible development and implementation of such technologies. Our work helps illustrate in an accessible manner the inner-workings of neural networks, a step towards aligning their use with responsible use. Researchers and practitioners must remain vigilant in addressing these challenges and ensuring that AI advances contribute positively to society. Additionally, interdisciplinary collaborations and public discussions can aid in raising awareness and developing robust strategies to mitigate potential risks.

