# OpenReview forum: "Neuron to Graph: Interpreting Language Model Neurons at Scale"
_ICLR.cc/2024/Conference — Submitted to ICLR 2024_

### Official Review · Reviewer_8BeY · 2023-10-21

**Soundness:** 2 fair
**Presentation:** 3 good
**Contribution:** 3 good
**Rating:** 6
**Confidence:** 3

**Summary:**

This paper attempts to explain each neuron with a graph representing sequences of tokens activating it. The approach consists of pruning, measuring importance by mapping salience of tokens to neurons, augmenting the data by replacing in the token with a masked language model, and building a graph to represent activating sequences. The sequences are empirically evaluated to show a middle ground between a precision-oriented baseline and a recall-oriented baseline.

**Strengths:**

- S1: This paper presents a novel method of explaining a neuron with a graph of activating sequences.
- S2: Similarly to some of the traditional explanation methods, this paper tries to understand neurons activities using a simpler model of tries, which simplified the evaluation and made the approach easier to understand.
- S3: The evaluation presents both strengths and weaknesses of the presented approach, as well as interesting trend between n-grams and precision/recall. In addition, they tried to explain activation trajectories and in-context learning using their approach.

**Weaknesses:**

- W1: The evaluation is done on only one target model, so it is hard to predict if this can generalize.
- W2: The paper makes some assumptions without strong justifications. For example, the role of the neuron is understood based on high activation values, but some neurons can have functions when its activation values are low. Also, the paper seems to assume that the token positions are carried all the way to the last layer, but the neurons at that position can represent a totally different composite notion than the input token especially in the later layers.
- W3: The performance is not great.

**Questions:**

- Q1: What are the examples producing token specific activations in the early layers, and activations from abstract concept in the later layers? How can we tell them if they are specific or abstract?
- Q2: The explanation of x-axis "Relative Token Index" in Figure 2 seems to be missing.

---

### Official Review · Reviewer_B6PG · 2023-11-01

**Soundness:** 3 good
**Presentation:** 3 good
**Contribution:** 3 good
**Rating:** 5
**Confidence:** 4

**Summary:**

This paper proposes a novel approach N2G to improve the interpretability of Large Language Models (LLMs).

**Strengths:**

1. The paper is well written.
2. N2G is scalable and can be used to interpret all neurons in a 6-layer Transformer model using a single Tesla T4 GPU. This makes it a practical tool for researchers to use.
3. N2G has been evaluated on two baseline methods and shown to be better at predicting neuron activation.
4. The generated graph representations can be flexibly used to facilitate further automation of interpretability research.

**Weaknesses:**

1. The description of the dataset Neuroscope is not clear.
2. The quantity evaluation is not sufficient. The major improvement is not significant. Lack of comparison with SOTA models.
3. The major contribution of interpretation part is not clear. The paper only mention the technology they used, not the performance the model achieved.

**Questions:**

1. Could you add more detail on the dataset?
2. Will other public well-known dataset suitable for this model?
3. What's the inspiration of this four-fold model?
4. Will the graph structure affect the final performance?

---

### Official Review · Reviewer_cE3E · 2023-11-01

**Soundness:** 2 fair
**Presentation:** 2 fair
**Contribution:** 2 fair
**Rating:** 1
**Confidence:** 4

**Summary:**

This paper proposes a method for summarizing highly activating dataset samples into a graph representation for each individual neuron. Given a neuron and a dataset, the top 20 most activating dataset samples are selected. Then from each sample a substring is selected such that the substring ends in the most activating token and the activation from this token given the substring is more than half the activation of the same token given the full string. The authors refer to this step as "pruning".

The pruned set is then augmented by replacing "salient" tokens in each sample with related tokens according to DistiBert. Saliency is identified by replacing each token with a padding token and thresholding the ration of maximum token activation before and after the replacement.

The resulting set of augmented substrings are then summarized in a Trie and visualized as a set of DAGs for each neuron. The Trie can be used to predict whether any given string is a highly activating string for a neuron.

**Strengths:**

A. The paper addresses an important problem: interpreting neurons in large language models, which can have implications for mechanistic interpretability, bias detection, and model safety.

B. The paper also provides a way to measure the quality of the generated neuron graphs by comparing them to the ground truth activations of neurons.

C. The paper is well-written and organized, with clear motivation, methodology, and experiments. It includes several figures and tables that illustrate the proposed method and its results.

**Weaknesses:**

A. No external validation of the method is done to show out of distribution generalization. Ground truth activation prediction is performed on the same model, and the same dataset that was used to create the Trie of highly activating substrings. This is a major limitation.

B. The authors acknowledge in the introduction that one problem in looking at highly activating samples in a dataset is that it provides an illusion of interpretability, but they do not address this problem. They augment the samples using related tokens, which does not introduce diversity in the dataset, and the interpretation is still limited to the distribution of data originally available in the dataset with some paraphrasing added. Even assuming the DistillBert augmentation does address this problem, this is not quantified in anyway and is not verifiable.

C. The proposed method involves several arbitrarily selected hyperparameters: top 20 activating samples for each neuron, pruning threshold, saliency threshold, augmentation threshold

D. There are no aggregate level results in the paper that actually help interpret the model. The paper could use an "applications" section where a behavior of interest in LLMs is studied and certain components of the model are discovered using the proposed method to be responsible for that behavior. On a related note,  in section 4.2 the jump from "repetition detection" to the claim of having discovered in-context-learning neurons is not supported by any evidence.

**Questions:**

Did the top 20 highly activating samples for each neuron satisfy a minimum activation level? Or is it possible that for some neurons the highly activating samples are actually not significantly activating?

---

### Official Review · Reviewer_QyST · 2023-11-01

**Soundness:** 2 fair
**Presentation:** 3 good
**Contribution:** 2 fair
**Rating:** 3
**Confidence:** 4

**Summary:**

The paper proposes an explanation method for large language models. Given a neuron of interest, the paper first identifies the text samples that cause high activation on the neuron. Then, through sample pruning, "pivot tokens" and their contexts within each sample are detected. After that, after augmentation, it identifies the tokens that are important for neuron activation on the pivot token. Finally, the graph is built by connecting the context tokens with the pivot token. The graph is regarded as the explanation for the neuron.

**Strengths:**

1. A timely topic of explaining large language models.
2. The proposed method can provide some useful information of LLM neurons. The pruning step and augmentation step make sense to me.
3. The method is intuitive and easy to follow.

**Weaknesses:**

1. Only using SoLU is an problem. We want to explain those models that are heavily used in practice, but not some testbeds.
2. The whole process is pretty ad-hoc. There is no rigorous definition of explanation. I would say this work is more like a post-hoc analysis, instead of research.
3. I can hardly call the result as a "graph", since it is just a pivot node with some context nodes. It IS a graph, but a very limited one.

**Questions:**

1. In the "pruning" part, how can we find $e_i$?
2. According to my understanding, the resultant graph is just a pivot node with some context nodes. Is this correct?
3. The paper title contains "at scale". How is this relected in the paper?

---

### Official Review · Reviewer_dCS6 · 2023-11-07

**Soundness:** 3 good
**Presentation:** 2 fair
**Contribution:** 3 good
**Rating:** 5
**Confidence:** 5

**Summary:**

This paper introduces a method named Neuron to Graph (N2G) to automatedly explain the behavior of neurons in LLM. This method constructs a token graph for each neuron according to the activation on training data examples. In experiment, using N2G, the author analyzes the neurons of a six-layer SoLU model which contains 18,432 neurons. Through validation, the author demonstrate that N2G is better at predicting neuron activation than token lookup and n-gram look up method. In addition, the author discovers some interesting pattern: some neuron are responsible for in-context learning, some neuron exhibits same behavior.

**Strengths:**

1. This paper proposes an automated method to interpret the behavior of neurons in language model by constructing a token tree. The visualization of token tree facilitates the interpretability of neurons and identification of neurons of interest.
2. The method can be easily scale to large language models.
3.  In experiment, the effectiveness of the method is validated by comparing with two other methods. In addition, the writing and presentation of this paper is good.

**Weaknesses:**

1. The paper mentions that the interpretability of neurons in deep layers is poor, but it does not provide any examples of poorly explained neurons.
2. This work is similar to [1], as both explore the behavior of neurons based on their activations to different tokens. This paper automates the interpretation of neurons using a graph, but does not significantly improve the interpretability of language model neurons. The differences with [1] need to be further explained in detail.
3. Current popular large language models use SwishGLU as the activation function, while this work focuses on models that use SoLU as the activation function. This creates a gap in the interpretability of large language models.

[1] Steven Bills, Nick Cammarata, Dan Mossing, Henk Tillman, Leo Gao, Gabriel Goh, Ilya Sutskever, Jan Leike, Jeff Wu, and William Saunders. Language models can explain neurons in language models. 2023.

**Questions:**

Why only take top 20 dataset examples? Are top 20 dataset examples enough to unravel the neuron behavior?

---

### Meta-Review · Area_Chair_7DKH · 2023-12-12

**Metareview:**

This paper investigates the inner mechanisms of large language models (LLM). Specifically, the authors introduce a method named Neuron to Graph (N2G) to explain the behavior of neurons in LLM. This method constructs a token graph for each neuron according to the activation on training data examples. The results demonstrate that N2G is better at predicting neuron activation than token lookup and n-gram look up method. However, the evaluation presented in this paper is not sufficient and convincing.

**Justification For Why Not Higher Score:**

The evaluation presented in this paper is not sufficient and convincing.

**Justification For Why Not Lower Score:**

N/A

---

### Decision · Program_Chairs · 2024-01-16

Reject